# DLLabelsCT: Annotation tool using deep transfer learning to assist in creating new datasets from abdominal computed tomography scans, case study: Pancreas

Henrik Mustonen[1]*, Antti Isosalo[1], Minna Nortunen[2,3], Mika Nevalainen[1,4], Miika T. Nieminen[1,4], Heikki Huhta[2,3]

1 Research Unit of Health Sciences and Technology, Faculty of Medicine, University of Oulu, Oulu, Finland, 2 Research Unit of Translational Medicine, Oulu University Hospital, Oulu, Finland, 3 Department of Surgery, Oulu University Hospital, Oulu, Finland, 4 Department of Diagnostic Radiology, Oulu University Hospital, Oulu, Finland

* henrik.mustonen@oulu.fi

**Data Availability Statement:** The annotation tool can be found from https://github.com/MIPT-Oulu/

## Abstract

The utilization of artificial intelligence (AI) is expanding significantly within medical research and, to some extent, in clinical practice. Deep learning (DL) applications, which use large convolutional neural networks (CNN), hold considerable potential, especially in optimizing radiological evaluations. However, training DL algorithms to clinical standards requires extensive datasets, and their processing is labor-intensive. In this study, we developed an annotation tool named DLLabelsCT that utilizes CNN models to accelerate the image analysis process. To validate DLLabelsCT, we trained a CNN model with a ResNet34 encoder and a UNet decoder to segment the pancreas on an open-access dataset and used the DL model to assist in annotating a local dataset, which was further used to refine the model. DLLabelsCT was also tested on two external testing datasets. The tool accelerates annotation by 3.4 times compared to a completely manual annotation method. Out of 3,715 CT scan slices in the testing datasets, 50% did not require editing when reviewing the segmentations made by the ResNet34-UNet model, and the mean and standard deviation of the Dice similarity coefficient was 0.82±0.24. DLLabelsCT is highly accurate and significantly saves time and resources. Furthermore, it can be easily modified to support other deep learning models for other organs, making it an efficient tool for future research involving larger datasets.

## Introduction

Artificial intelligence (AI) has shown promising results in the field of medical research and medical image analysis [1]. Deep learning (DL) is a subset of AI that involves the use of artificial neural networks to learn and recognize patterns in data, such as computed tomography (CT) and magnetic resonance imaging (MRI) scans, which can provide detailed information

DLLabelsCT Any patients related data is not available due to Finnish legislation.

**Funding:** HH: (grant no. 00210395) Finnish Culture Foundation https://skr.fi/en, (grant no. 202210072) Mary and George C. Ehrnroot Foundation https://marygeorg.fi/en/home/, (grant no. 5785) Finnish Medical Foundation https://laaketieteensaatio.fi/en/home/, Sigrid Jusélius Foundation https://www.sigridjuselius.fi/en/ AI: (grant no. 10221743) Finnish Culture Foundation https://skr.fi/en, Jane and Aatos Erkko Foundation https://jaes.fi/en/frontpage/, Technology Industries of Finland Centennial Foundation https://techfinland100.fi/en/, (grant no. 220106) Wihuri Foundation https://wihurinrahasto.fi/?lang=en MTN: Jane and Aatos Erkko Foundation https://jaes.fi/en/frontpage/, Technology Industries of Finland Centennial Foundation https://techfinland100.fi/en/ None of the funding sources had no involvement in study design; in the collection, analysis and interpretation of data; in the writing of the report; and in the decision to submit the article for publication.

**Competing interests:** The authors have declared that no competing interests exist.

about the abdominal organs or location and characteristics of tumors [2, 3]. To enhance the specificity and sensitivity of the DL algorithms for instance in cancer detection, it is crucial to have access to high-quality data representing different cancer types and stages [2]. This requires large datasets of medical images, which are annotated by expert radiologists, vital for DL algorithms to learn accurately. The task is time-consuming and costly [2]. Annotation tools typically provide semi-automated segmentation methods, which are either based on previous annotations requiring input from the user, like selecting the region for the automatic segmentation method [4]. Fully automated segmentation methods do not necessarily require user input and with large datasets it would improve efficiency and be beneficial.

Fully automated methods are the most effective when the segmented object differs heavily from its surroundings [5]. In CT scans the surrounding organs have similar Hounsfield Unit (HU) -values, making automatic segmenting more difficult [5]. DL-based segmentation methods have proven very accurate for segmenting organs in abdominal CT scans [5]. There are only a few validated annotation tools that support DL-methods, such as RIL-Contour, which supports deep learning models using Keras running on Tensorflow [6]. However, Philbrick et al. concentrated primarily on the mechanical aspects of the annotation tool, rather than illustrating how a deep learning model could be trained to aid in the annotation process [6]. Training of DL algorithms to clinical standards requires extensive datasets, and their processing is exceedingly time-consuming. Tools that automatically annotate organs or tumors from CT images can expedite the management of these large datasets. Yet, the availability of validated and published DL-based annotation tools remains limited.

The purpose of this study was to evaluate the efficacy of our newly developed deep learning-based annotation tool, named DLLabelsCT (Deep Learning Labels Computed Tomography), in accelerating the annotation process of organs in CT scans versus traditional manual annotation. DLLabelsCT is available from https://zenodo.org/records/10226990. The pancreas was selected as the organ for annotation, since its segmentation is a difficult task, due to the varying of the parenchymal shape, density, contrast enhancement and size within the abdominal CT-scan [7] making it a more challenging object for DL compared, for instance, to the liver.

## Materials and methods

### Datasets

Four independent CT scan datasets were used. The CT imaging in these datasets was performed in the portal venous phase and the image size was 512-by-512 pixels with varying pixel spacings. The scans were originally in Digital Imaging and Communications in Medicine (DICOM) format. Windowing was performed on DICOM-images, with the window width $w_l$ of 400 HU and the window center $w_c$ of 50 HU, for improved contrast between the tissue types. Windowing sets the images output values based on the following equation,

$$y = \left( \frac{x - (w_c - 0.5)}{w_l - 1} + 0.5 \right) \cdot (y_{max} - y_{min}) + y_{min} \tag{1}$$

where $x$ is the input value, $y$ is the output value and $y_{min}$ and $y_{max}$ are the minimum and maximum possible outputs for the image format, i.e. for 16-bit images the values are 0 and 65535. Values outside the window were set to the maximum and minimum. The individual slices were then transformed into 16-bit PNG images from the DICOM format. The PNG images were then used to train the convolutional neural network (CNN) model.

**Training dataset.** To train the DL-based segmentation method described in this study, an open-access National Institutes of Health Clinical Center dataset of contrast enhanced CT

scans from the Cancer Imaging Archive was utilized [8–10]. The dataset consisted of 80 abdominal CT scans from healthy subjects with annotations for the pancreas. The dataset is referred to as Pancreas-CT from hereafter. The dataset contained a total of 18,942 axial slices and the slice thickness of the scans was 1.5–2.5 mm. This dataset was divided into training and testing datasets, with the testing dataset containing 20% of the scans. The patients were randomly split into the datasets without any stratification. According to the Cancer Imaging Archive, the volumes were acquired using CT scanning systems by Philips (Amsterdam, Netherlands) and Siemens (Erlangen, Germany). The X-ray tube voltages of the scanners were 120 kVp. This dataset was initially downloaded in November 2022.

**Validation dataset.** A dataset from the Oulu University Hospital was used to further train the segmentation model and to validate our novel in-house annotation tool. We refer to this dataset later as the Oulu validation dataset. The Oulu validation dataset contained CT scans from 606 patients, 313 with healthy pancreas, 218 with cancerous pancreas and 75 with other pancreatic diseases, such as intraductal pancreatic mucinous neoplasms, chronic pancreatitis etc. The slice thickness in Oulu validation dataset's scans varied between 0.625–7 mm. These scans contained a total of 96,429 axial slices. The CT scanners were manufactured by Toshiba (Minato, Tokyo, Japan), Philips (Amsterdam, Netherlands), Canon Medical Systems (Otawara, Tochigi, Japan), Siemens (Erlangen, Germany) and GE Medical Systems (Chicago, Illinois, United States). Tube voltages ranged between 80–140 kVp. This dataset was accessed and annotated between January and March 2023.

**Testing datasets.** Two accessory datasets from the Kuopio University Hospital (CT scans of 56 patients) and from Turku University Hospital (CT scans of eight patients) were combined to form the external testing dataset for accessing the performance of the annotation tools. All CT scans were of patients with pancreatic cancer. The dataset had a total of 26,154 axial slices, of which 12,227 were in the wanted portal venous phase. The slice thickness in these scans was 1–5 mm. The CT scanners were made by Toshiba (Minato, Tokyo, Japan), GE Medical Systems (Chicago, Illinois, United States) and Siemens (Erlangen, Germany). Tube voltages were 80–120 kVp. These datasets were accessed and annotated between April and May 2023.

The second testing dataset was from the Oulu University Hospital, containing 70 CT-scans with pancreatic cancer and which had not been previously used in the training dataset. We refer to this dataset later as the Oulu testing dataset. This dataset had a total of 17,558 axial slices with slice thickness of 0.5–5 mm. The CT scanners were made by Toshiba (Minato, Tokyo, Japan), GE Medical Systems (Chicago, Illinois, United States), Siemens (Erlangen, Germany), Philips (Amsterdam, Netherlands) and Canon Medical Systems (Otawara, Tochigi, Japan). Tube voltages were 80–120 kVp. The validation datasets were divided into two, with each half annotated by using a different tool (see chapters *Annotation tools* and *Method evaluation)*. This testing dataset was accessed and annotated after the Oulu validation dataset, between March and April 2023.

## Deep learning model

A 2-D CNN model with an encoder-decoder architecture was trained and used to segment pancreas from the CT-scans. The architecture utilizes a downsampling encoder that provides features for the upsampling decoder, which provides a prediction for each pixel in the image (Fig 1). The model employs a ResNet34 [11] encoder that had been pretrained on the ImageNet [12] dataset, combined with a U–Net [13] decoder with randomly initialized weights. We refer to the model later as ResNet34UNet. This model was used because it provided good results on breast mass segmentation in our previous study [14].

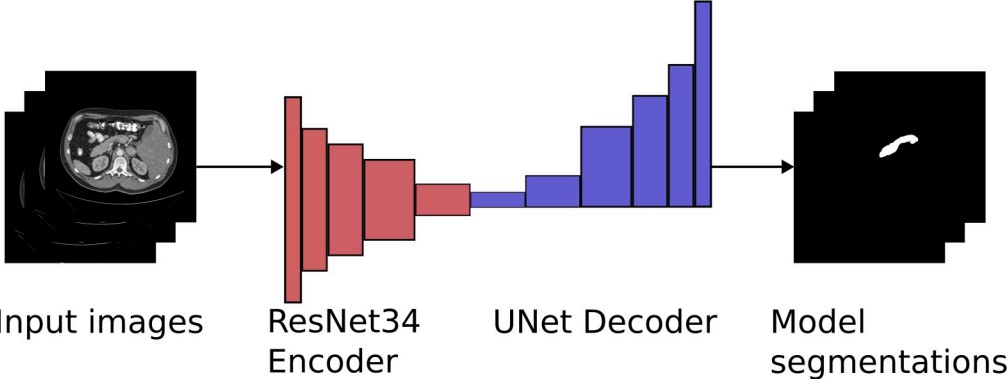

**Fig 1. Example of how the Resnet34UNet model functions.**

**Model training.** Initial training of the ResNet34UNet model was conducted on the open Pancreas-CT dataset. The training was implemented in a fully supervised manner by using the pixel-accurate annotations provided with the dataset. The training was performed on the training part of the Pancreas-CT dataset, including slices that did not contain pancreas.

Five-fold-cross-validation with non-interlapping groups was used, with the groups generated with Scikit-learn's [15] GroupKFold function. The model was trained for 50 epochs with a batch size of 16. The loss function utilized during training was Focal Tversky loss (FT) [16, 17], defined as

$$ FT = \left( \frac{TP + 1}{TP + \alpha \cdot FP + \beta \cdot FN + 1} \right)^{\gamma} \tag{2} $$

where TP is the number of true positive predictions, FP is the number of false positive predictions and FN is the number of false negative predictions, the $\alpha$ parameter is 0.7, $\beta$ is 0.3 and $\gamma$ is 0.75. Multiple data augmentations were used during training (Table 1). The augmentations were retrieved from the Streaming Over Lightweight Transformations (SOLT) [18] library version 0.1.8 and each of the augmentations had a 50% chance of occurring. Adam [19] was used as the optimizer during the training, with a multi-step learning rate scheduler. The initial

**Table 1. Data augmentations used on the data during model training.**

| Augmentation | Parameters [units] |
|---|---|
| Rotation | [−5, 5] [rad] |
| Scale | [0.6, 1.4] [au] |
| Translation | 80 [px] |
| Random crop | 448 × 448 [px] |
| Horizontal and vertical flip | [0.5, 1.8] [au] |
| Gamma correction | |
| Brightness and contrast | [30, 100] [au] |
| Salt and pepper noise | 0.1 [au] |
| Gaussian noise | 0.5 [au] |
| Gaussian blur (kernel sizes, sigma) | (3, 7, 11), (1, 10) |
| Median blur (kernel sizes, sigma) | (3, 7, 11), (1, 10) |
| Cutout | 20% |

Rad: radian, au:arbitrary unit, px: pixels

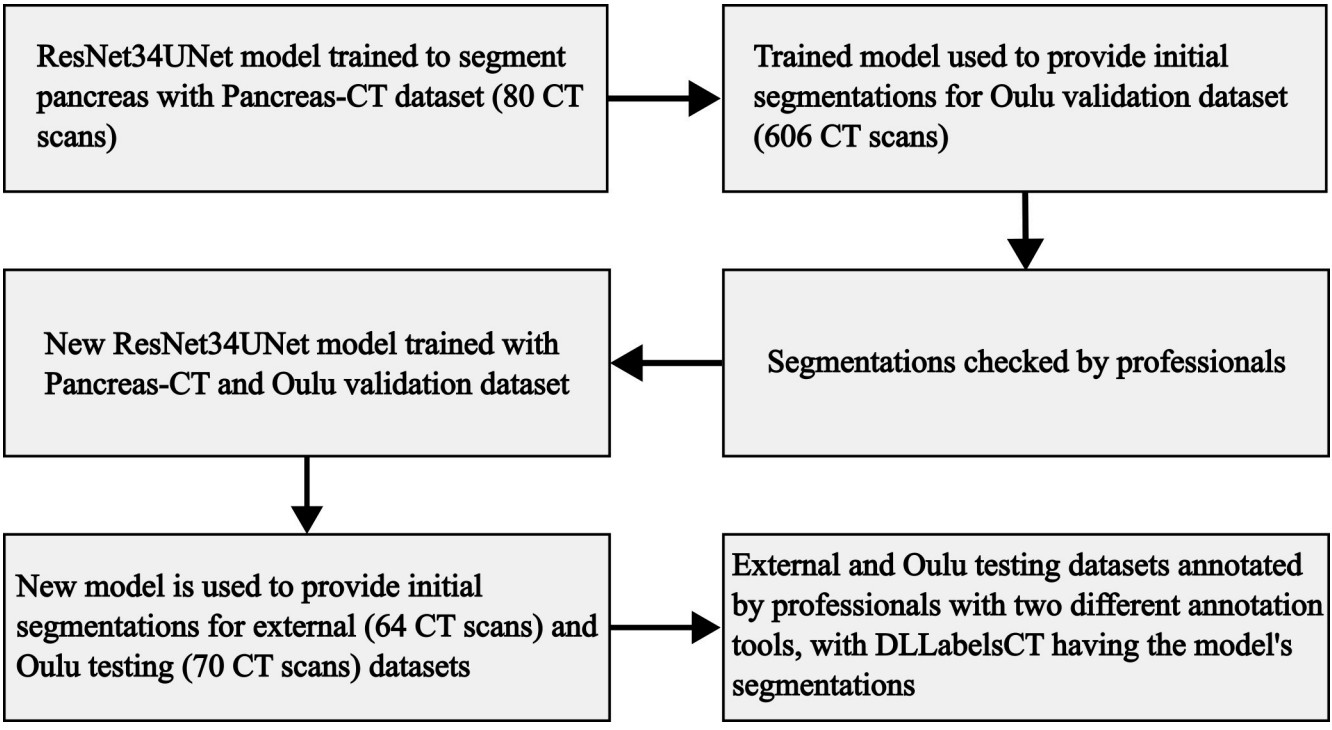

**Fig 2. Flow chart showing the model training process.**

learning rate and weight decay were set at $10^{-4}$. The learning rate was reduced by a factor of 0.1 after 20, 30 and 40 epochs. The computer used had a Nvidia GeForce RTX 3090 graphics processing unit (GPU), an AMD Ryzen 9 5950 16-core central processing unit (CPU) and 64 GB of 2400 MHz random-access memory (RAM). The computer's OS was Ubuntu 20.04.1 and the used Python version was 3.10.10 with PyTorch [20] 1.13.1 and CUDA version 11.6. With this hardware the initial training took 24 hours.

After training the ResNet34UNet model, it was used to provide initial annotations for the Oulu validation dataset. These initial annotations were then reviewed by our medical professionals (H.H, M.N and M.N). This checked segmented dataset was then combined with the Pancreas-CT dataset to train a new ResNet34Unet model, using the same training arguments as the initial model. To reduce training time, the number of images in the combined dataset was reduced by removing slices not containing pancreas. The training dataset contained 31,634 slices. The model training and application process is summarized in (Fig 2).

## Annotation tools

Two different annotation tools were used and compared in the study. Our novel in-house tool DLLabelsCT was developed for the current study using Python (version 3.10.10), which uses DL to assist in annotating (Fig 3). Additionally a previously developed MATLAB-based (2020a, Natick, MA, United States) tool named MammogramAnnotationTool [21] without any intelligent features (Fig 4) was used. This tool was modified to support CT scans and was initially used in annotating the datasets, the modified tool was renamed CTAnnotationTool. DLLabelsCT was developed to speed up the annotation process. DLLabelsCT combines segmenting CT scans with a PyTorch based CNN and annotating the images manually. It uses a PyQt (2022, Dorchester, United Kingdom) based interface for the annotating. PyQt was

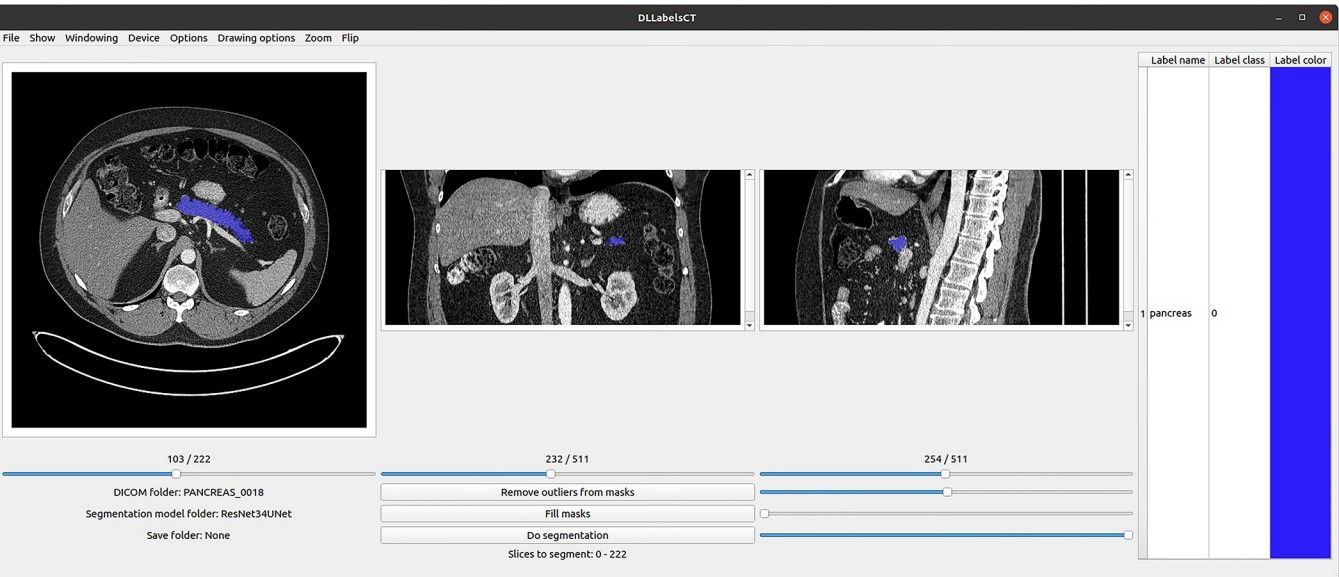

**Fig 3. DLLabelsCT with an example scan from the Pancreas-CT dataset.** The shown masks can be modified, and new labels added from the menu on the right.

selected as the base since it uses Python, as does PyTorch, and it is simple to use and understand. DLLabelsCT saves the annotations and the individual axial slices as PNG images, which can then directly be used as data in our DL model's training pipeline. The tool supports

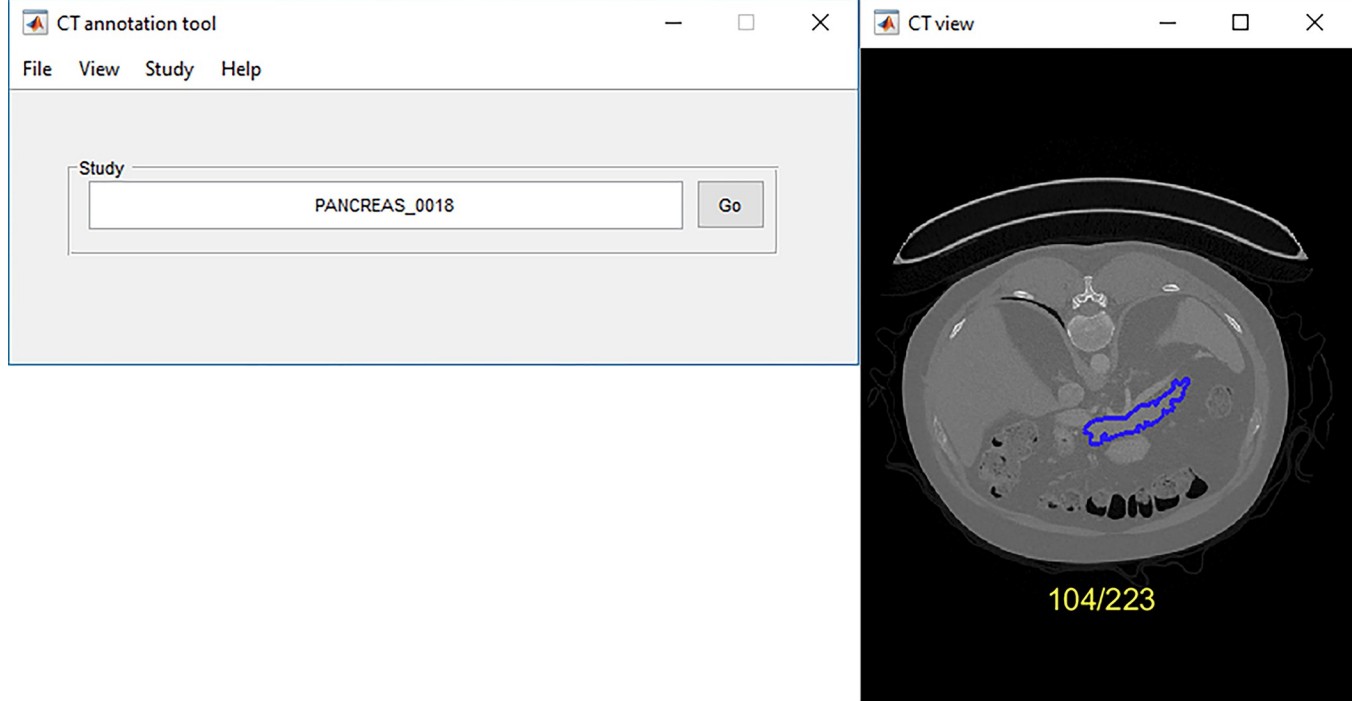

**Fig 4. MATLAB-based annotation tool graphical user interface with an example scan from the Pancreas-CT dataset.** The study can be changed from the window on the left and labels can be added on the CT scan on the window to the right.

segmentation models with ResNet encoders and UNet or Feature Pyramid Network (FPN) [22] decoders.

## Method evaluation

The external testing dataset and the Oulu testing dataset were used for testing the accelerating effect of our fully automated segmentation tool. Experienced, independent, and mutually blinded pancreatic surgeons (H.H for external and M.N. for Oulu) provided annotations using both the conventional CTAnnotationTool and the novel fully automated DLLabelsCT. The CTAnnotationTool was used to annotate 32 patients' pancreas from the external testing dataset and 31 patients from the Oulu testing dataset. The rest of the patients, 32 from the external testing dataset and 39 from the Oulu testing dataset, were annotated using DLLabelsCT with the initial DL model segmentations. Initial segmentations review was done with similar method between reviewers; segmentation needs to fill the entire pancreas and if not, it was corrected. The number of annotated slices and the time required for annotation were recorded for all patients. Additionally, we assessed the extent of revisions needed to the segmentations made by the ResNet34Unet model. The Dice similarity coefficient (DSC), defined as

$$\text{DSC} = \frac{2TP}{2TP + FP + FN} \tag{3}$$

where TP is the number of true positive predictions, FP is the number of false positive predictions and FN is the number of false negative predictions, was calculated between the boolean masks made by the model and the segmentations modified with the fully automated segmentation tool.

## Ethical statement

This study complied with the ethical standards of the Oulu University Institutional Research Committee and the Declaration of Helsinki (as revised in 2013). Data collection from included hospitals was approved by the Finnish Social and Health Data Permit Authority (FINDATA, Dnro THL/3606/14.02.00/2020). Individual patient consent for this retrospective analysis was waived. Additionally, ethical permission from the local wellbeing services county (Pohde) and ethical permission EETMK: 81 / 2008 were received.

## Results

ResNet34UNet model trained on the training subset of Pancreas-CT had lower segmentation accuracies than the ResNet34UNet model trained on the Oulu validation dataset and the training subset of Pancreas-CT (Table 2, Fig 5). The segmentation accuracy was higher on the external testing dataset than the Oulu testing dataset (Table 2). The pancreas detection results (whether the model's segmentation mask and the final annotation mask overlap) for each

**Table 2. Segmentation results of the ResNet34UNet model trained on the Pancreas-CT dataset (Model 1) and the ResNet34UNet model trained on Pancreas-CT and Oulu validation dataset (Model 2).** Model 2 was used in segmenting the external testing dataset and Oulu testing dataset. Results presented as mean±standard deviation.

|  | Model 1 on Pancreas-CT | Model 2 on Pancreas-CT | External testing dataset | Oulu testing dataset |
|---|---|---|---|---|
| DSC | 0.79±0.14 | 0.82±0.14 | 0.89±0.21 | 0.80±0.24 |
| Recall | 0.75±0.18 | 0.80±0.17 | 0.89±0.23 | 0.80±0.27 |
| Precision | 0.88±0.13 | 0.87±0.13 | 0.94±0.17 | 0.84±0.22 |

DSC: Dice similarity coefficient

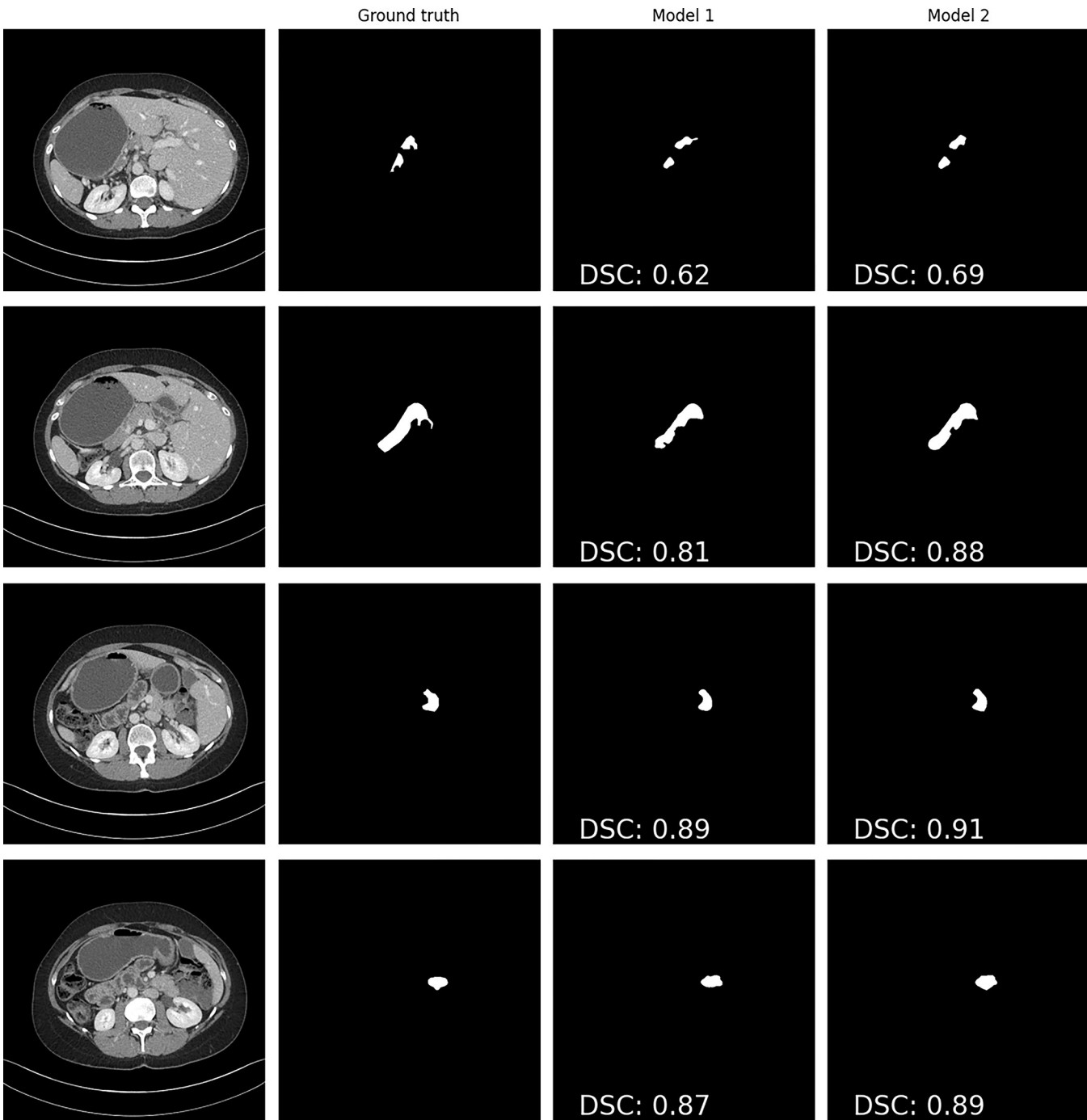

**Fig 5. Example result on a CT scan from the Pancreas-CT dataset.** Model 1 is the ResNet34UNet model trained on the training subset of the Pancreas-CT dataset and Model 2 is the ResNet34UNet model trained on both the training subset of the Pancreas-CT dataset and Oulu validation dataset.

individual slice in the CT scan were high for both of the models, with the ResNet34UNet model trained on the training subset of Pancreas-CT and the Oulu validation dataset having the higher accuracy on the testing subset of Pancreas-CT, and the accuracy was higher on the external testing dataset than the Oulu testing dataset (Table 3).

**Table 3. Pancreas detection (whether the ground truth mask and the model segmentation overlap on a CT scan slice) results of the ResNet34UNet model trained on the Pancreas-CT dataset (Model 1) and the ResNet34UNet model trained on Pancreas-CT and Oulu validation dataset (Model 2).** Model 2 was used in segmenting the external testing dataset and Oulu testing dataset. Results presented as mean±standard deviation.

|  | Model 1 on Pancreas-CT | Model 2 on Pancreas-CT | External testing dataset | Oulu testing dataset |
|---|---|---|---|---|
| Accuracy | 95.2% | 95.4% | 97.6% | 96.5% |
| Specificity | 98.8% | 96.3% | 98.3% | 98.9% |
| Sensitivity | 88.9% | 93.8% | 94.9% | 88.3% |
| PPV | 97.7% | 93.6% | 94.1% | 96.0% |
| NPV | 93.9% | 96.4% | 98.6% | 96.6% |

PPV: Positive predictive value, NPV: Negative predictive value

Annotating the datasets was faster with DLLabelsCT than with CTAnnotationTool, even though the dataset annotated with DLLabelsCT contained more slices (Tables 4 and 5). The ResNet34UNet model's segmentations made annotating 3.4 times faster on average (Tables 4 and 5). In the external testing dataset 62% of the model segmentations were accepted without revision, while 44% of the segmentations in Oulu testing dataset needed no adjusting. In total 50% of the segmentations did not need adjusting.

## Discussion

The current study demonstrates that it is possible to create a highly accurate DL- based fully automated segmentation tool for annotating organs from abdominal CT scans with a relatively small amount of data. Furthermore, we show that the use of fully automated software saves a significant amount of time, making it cost-effective.

The used DL segmentation method has higher DSC than the initial method proposed with the Pancreas-CT dataset [7], with the ResNet34UNet model having a mean DSC of 0.82 over the mean DSC of 0.72 of the method proposed in the previously mentioned article. Newer DL-based segmentation methods have achieved a larger DSC than our method, with a DL segmentation method achieving a mean DSC of 0.90 on the Pancreas-CT dataset [23]. The DL method used in our study is more of an example of the potential of DL and DLLabelsCT can be easily modified to support different PyTorch-based segmentation models.

DL neural networks are under intensive research and have a wide range of possible applications also in the field of medical research. For instance, genomics, metagenomics, histological and radiological image recognition benefit from AI evolution [23]. Recently various studies have shown promising results from DL network for pancreas segmentation. However, prior DL-based pancreatic segmentation studies utilized the Pancreas-CT dataset (n = 82). While the previously suggested DL networks attained commendable performance for pancreas segmentation (mean DSC of 0.866 and 0.854), the available data remains insufficient to establish

**Table 4. Annotation times for the external testing dataset.**

|  | Python DLLabelsCT | MATLAB CTAnnotationTool |
|---|---|---|
| Number of scans | 32 | 32 |
| Total time (seconds) | 3223 | 8486 |
| Total slices | 1313 | 926 |
| Time per slice (seconds) | 2.5 | 9.2 |
| Average time per scan (seconds) | 94.8 | 73.0 |
| Average number of slices in a scan | 38.6 | 27.2 |

**Table 5. Annotation times for the Oulu testing dataset.**

|  | Python DLLabelsCT | MATLAB CTAnnotationTool |
|---|---|---|
| Number of scans | 39 | 31 |
| Total time (seconds) | 10304 | 12287 |
| Total slices | 2402 | 884 |
| Time per slice (seconds) | 4.3 | 13.9 |
| Average time per scan (seconds) | 264.2 | 396.4 |
| Average number of slices in a scan | 61.6 | 28.5 |

the reliability of these networks since DL-based medical image segmentation is highly dependent on the number of data points [24, 25].

To the best of our knowledge, the only study conducted with a relatively large amount of data (1006 patients) for normal healthy pancreatic segmentation is the Korean study, whose results closely aligned with our own, DSC (0.84 and our 0.82) [26]. It's important to note, however, that our dataset included 43% of patients with pathological pancreases. Additionally, our segmentation method for the pancreas achieved a DSC of 0.80 within the Oulu testing dataset, which exclusively comprises patients diagnosed with pancreatic ductal adenocarcinoma which is significantly impacting to pancreas normal shape, size and volume of pancreatic parenchyma [27]. Previous studies have shown more modest performance in similar conditions [28]. Promising results have also been achieved using neural networks for cancer detection and disease prognosis classification [29, 30]. Using CNN, small pancreatic cancers with a diameter of less than 2cm were identified from CT images with the same or even better accuracy than radiology specialists [31].

DL methods can give more reliable outputs than other segmentation methods, such as active contours, region growing, histogram-based methods etc. since the DL models learn the varying shapes and contrast of the intended target [32]. DL methods do not require any additional input from the user and the initial annotation masks can be created with no imminent user. The disadvantages of DL are the long training time and the large computational requirements.

Developing DL applications for pre-operative assessment for instance, of pancreatic tumors requires large patient cohorts to capture disease variations [33]. Challenges in developing algorithms from large nation-wide datasets include varying imaging equipment, contrast agent concentrations, and slice thicknesses [34]. The algorithm must also learn patient-specific factors such as comorbidities, age, body composition, and circulatory issues to function effectively [34]. Given resource limitations, manual annotation of massive datasets is unfeasible. Automated DL annotation tools like DLLabelsCT are necessary for processing these datasets. Validation, reporting, and publication of these tools are crucial for assessing the quality and reliability of the data processed. In our study, the in-house tool DLLabelsCT was 3.4 times faster than manual annotation, even with pancreatic cancer cases.

There are some limitations to this study. First, the small amount of CT scans used in evaluating differences between the annotation tools causes bias and could hamper the comparison. This was compensated for by having data from multiple sources with different types of data, which contributes to a better generalizability of the developed model. Similarly, different data from various CT scanners and sources were used in training the DL model. A technical limitation arises from using DL with a reasonable speed which requires a compatible GPU. Nonetheless, DLLabelsCT can be used without a GPU and the model's segmentation masks can be provided from a separate computer with a GPU.

## Conclusion

The results demonstrate that our DL based fully automated segmentation and annotation tool DLLabelsCT for pancreas is highly accurate and saves time and resources significantly. Moreover, it could easily modify to detect other organs as well with small data amount and will be an efficient tool for future research with larger datasets. The annotation tool is publicly available at https://zenodo.org/doi/10.5281/zenodo.10226989

## Acknowledgments

Special thanks: Esa Liukkonen and Anne Kukkonen for their excellent technical assistance on data facilitation

## Author Contributions

**Conceptualization:** Henrik Mustonen, Antti Isosalo, Minna Nortunen, Mika Nevalainen, Miika T. Nieminen, Heikki Huhta.

**Data curation:** Henrik Mustonen, Heikki Huhta.

**Formal analysis:** Henrik Mustonen, Antti Isosalo, Minna Nortunen, Mika Nevalainen, Heikki Huhta.

**Funding acquisition:** Heikki Huhta.

**Investigation:** Henrik Mustonen, Antti Isosalo, Minna Nortunen, Mika Nevalainen, Heikki Huhta.

**Methodology:** Henrik Mustonen, Antti Isosalo, Heikki Huhta.

**Project administration:** Henrik Mustonen, Heikki Huhta.

**Software:** Henrik Mustonen, Antti Isosalo.

**Supervision:** Henrik Mustonen, Miika T. Nieminen, Heikki Huhta.

**Validation:** Henrik Mustonen, Antti Isosalo, Minna Nortunen, Mika Nevalainen, Heikki Huhta.

**Visualization:** Henrik Mustonen.

**Writing – original draft:** Henrik Mustonen, Heikki Huhta.

**Writing – review & editing:** Henrik Mustonen, Antti Isosalo, Minna Nortunen, Mika Nevalainen, Miika T. Nieminen, Heikki Huhta.

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
