## [Decision Letter · Decision Letter 0]

7 Aug 2024

PONE-D-24-21753DLLabelsCT: Annotation tool using deep transfer learning to assist in creating new datasets from abdominal computed tomography scans, case study: pancreasPLOS ONE

Dear Dr. Mustonen,

Thank you for submitting your manuscript to PLOS ONE. After careful consideration, we feel that it has merit but does not fully meet PLOS ONE’s publication criteria as it currently stands. Therefore, we invite you to submit a revised version of the manuscript that addresses the points raised during the review process.

We look forward to receiving your revised manuscript.

Kind regards,

Hadeel K. Aljobouri

Academic Editor

PLOS ONE

Journal Requirements:

Additional Editor Comments:

The conclusion section must be added and separated from the discussion section.

Reviewers' comments:

Reviewer's Responses to Questions

**Comments to the Author**

1. Is the manuscript technically sound, and do the data support the conclusions?

Reviewer #1: Yes

Reviewer #2: Yes

2. Has the statistical analysis been performed appropriately and rigorously? 

Reviewer #1: Yes

Reviewer #2: Yes

3. Have the authors made all data underlying the findings in their manuscript fully available?

Reviewer #1: Yes

Reviewer #2: Yes

4. Is the manuscript presented in an intelligible fashion and written in standard English?

Reviewer #1: Yes

Reviewer #2: Yes

5. Review Comments to the Author

Reviewer #1: This study introduces an annotation tool utilizing convolutional neural networks to enhance image analysis in medical research called DLLabelsCT. It aims to significantly reduce the time and labor needed for annotating medical images. The authors used pancreas CT scans as data. The study demonstrates DLLabelsCT's speed and accuracy, making it valuable for large datasets and adaptable for different organs. The authors compared their results with a modified MATLAB-based CTAnnotationTool, showing that DLLabelsCT is accurate, time-efficient, and resource-saving. Additionally, the authors provided all necessary libraries and packages, ensuring reproducibility for future research. This study is very impressive. I have a couple of comments:

• The authors mentioned that in literature there are few annotation tools that support DL methods (Introduction second paragraph). It would be helpful to include more information about the existing methods, such as the models they used, their performance, and the size and type of datasets they trained on.

• A flow chart of the whole process would be very useful, especially for understanding the datasets used. For instance, the authors have training, validation, and testing datasets consisting of different sources. Creating a visualization that shows which data was used for training before medical professionals' review, and which dataset was used afterward, would make the study much clearer.

• The comparison with CTAnnotationTool regarding time and resources is very useful (Table 4, and 5). However, it would be beneficial to include performance metrics for CTAnnotationTool such as recall and precision. This would provide a clearer demonstration of DLLabelsCT's superior speed and also performance compared to CTAnnotationTool.

Reviewer #2: dear authers

in the section (materials) you mentioneed the chosen image size of 512 by 512, it is advised to provide in short the logic behind this size.

in the section (training dataset) the scanning systems mentioned in the study have different manufacturer, what effect does this has on the aqcuired data and how does this eventually affect the model?

in the section (validation dataset) the dataset from Oulu university was explained stating that 218 scans were cancerous, what methods of diagnosis were used?

in the section (model training) it is mentioned that augmentation was used, how did this change the results? in the same section it is specified that a number of non pancrease containing slices were eleminated to enhance the time factor, could you please specify the amount in time redcuction if possible, what was the time consumed before eliminating the slices and what time was consumed after?

6. PLOS authors have the option to publish the peer review history of their article (what does this mean?). If published, this will include your full peer review and any attached files.

Reviewer #1: No

Reviewer #2: **Yes: **Aya Abdul Salam Alsalihi

---

## [Author Response · Author response to Decision Letter 0]

18 Sep 2024

Academic editor’s comments: 

Journal Requirements: 

Author response: Thank you for your comment, the manuscript will be changed to meet these requirements. 

Author response: We agree with this, and have already made the annotation tool’s code open-source and available in https://github.com/MIPT-Oulu/DLLabelsCT

Author response: There seems to have been a misunderstanding during the initial submission, this will be fixed in the resubmission. 

Additional Editor Comments: 

The conclusion section must be added and separated from the discussion section. 

 Author response: This will be done in the resubmission. 

Reviewer #1: This study introduces an annotation tool utilizing convolutional neural networks to enhance image analysis in medical research called DLLabelsCT. It aims to significantly reduce the time and labor needed for annotating medical images. The authors used pancreas CT scans as data. The study demonstrates DLLabelsCT's speed and accuracy, making it valuable for large datasets and adaptable for different organs. The authors compared their results with a modified MATLAB-based CTAnnotationTool, showing that DLLabelsCT is accurate, time-efficient, and resource-saving. Additionally, the authors provided all necessary libraries and packages, ensuring reproducibility for future research. This study is very impressive. I have a couple of comments: 

• The authors mentioned that in literature there are few annotation tools that support DL methods (Introduction second paragraph). It would be helpful to include more information about the existing methods, such as the models they used, their performance, and the size and type of datasets they trained on. 

Author response: It is true that our article does not give much information about other annotation tools with deep learning capabilities. However, the articles about those tools focused more on the tool itself, instead of the deep learning model and their performance, meaning that such a comparison is not possible. Our manuscript focuses more on showing that a deep learning model can be trained to assist in annotating with a small open access dataset (Pancreas-CT in our case). 

Following changes were made to text, page 3 introduction section: “However, Philbrick et al. concentrated primarily on the mechanical aspects of the annotation tool, rather than illustrating how a deep learning model could be trained to aid in the annotation process.” 

 • A flow chart of the whole process would be very useful, especially for understanding the datasets used. For instance, the authors have training, validation, and testing datasets consisting of different sources. Creating a visualization that shows which data was used for training before medical professionals' review, and which dataset was used afterward, would make the study much clearer. 

Author response: Thank you for your comment, we agree that such a flow chart would be useful and have added it into the manuscript as Figure 2 

 • The comparison with CTAnnotationTool regarding time and resources is very useful (Table 4, and 5). However, it would be beneficial to include performance metrics for CTAnnotationTool such as recall and precision. This would provide a clearer demonstration of DLLabelsCT's superior speed and also performance compared to CTAnnotationTool. 

Author response: We agree with the referee that including metrics for the CTAnnotationTool would be demonstrative. However, since the CTAnnotationTool lacks machine learning capabilities, it is not feasible to provide such performance metrics. 

Reviewer #2: dear authers 

 in the section (materials) you mentioneed the chosen image size of 512 by 512, it is advised to provide in short the logic behind this size. 

Author response: Thank you for your comment, this image size is the size of the CT –scans, there were no other options. 

 in the section (training dataset) the scanning systems mentioned in the study have different manufacturer, what effect does this has on the aqcuired data and how does this eventually affect the model? 

 Author response: Referee pointed out an excellent detail. Generally, CT scans taken with different CT scanners provide some information to the model about how varied the imaging can be, with different pixel spacings and slight variations in grayscale values, etc. This eventually leads to the model better responding to those variations, increasing the segmentation quality. 

 in the section (validation dataset) the dataset from Oulu university was explained stating that 218 scans were cancerous, what methods of diagnosis were used? 

Author response: Thank you for your comment, all scans from the patients with cancer imaging were later treated with surgery, and the diagnosis was based on the pathological examination of surgical specimens 

 in the section (model training) it is mentioned that augmentation was used, how did this change the results? 

Author response: This is a good comment, augmentations have been proven to be an effective way to reduce model overfitting during training, especially on smaller datasets. This means that the model responds better to variations in the dataset and the model provides better quality segmentations on images that it has not seen. 

in the same section it is specified that a number of non pancrease containing slices were eleminated to enhance the time factor, could you please specify the amount in time redcuction if possible, what was the time consumed before eliminating the slices and what time was consumed after? 

Author response: Thank you for your comment, the datasets (Pancreas-CT and Oulu validation dataset) contained a total of 115,371 axial slices. The initial training dataset contained 18,942 axial slices and the model training took 24 hours. If all of those 115,371 axial slices were used during training, the model training would take around a week. This would’ve been too long for us, which means that the number of slices had to be reduced.

---

## [Editor Report · Decision Letter 1]

2 Oct 2024

PONE-D-24-21753R1DLLabelsCT: Annotation tool using deep transfer learning to assist in creating new datasets from abdominal computed tomography scans, case study: pancreasPLOS ONE

Dear Dr. Mustonen,

Thank you for submitting your manuscript to PLOS ONE. After careful consideration, we feel that it has merit but does not fully meet PLOS ONE’s publication criteria as it currently stands. Therefore, we invite you to submit a revised version of the manuscript that addresses the points raised during the review process.

We look forward to receiving your revised manuscript.

Kind regards,

Hadeel K. Aljobouri

Academic Editor

PLOS ONE

Journal Requirements:

Additional Editor Comments:

The number of references is very small, we prefer to add more. I recommend the authors to read these papers:

A. A. Alsalihi, H. K. Aljobouri, and E. A. K. ALTameemi, “GLCM and CNN Deep Learning Model for Improved MRI Breast Tumors Detection,” International Journal of Online and Biomedical Engineering (iJOE), vol. 18, no. 12, pp. 123–137, Sep. 2022, doi: 10.3991/IJOE.V18I12.31897.

J. Yahya Rbat, H. K. Aljobouri, and A. M. Hasan, “MRI brain tumor classification using robust Convolutional Neural Network CNN approach,” pp. 258–261, Jan. 2023, doi: 10.1109/IICCIT55816.2022.10009913.

N. H. Alkurdy, H. K. Aljobouri, and Z. K. Wadi, “Ultrasound renal stone diagnosis based on convolutional neural network and VGG16 features,” International Journal of Electrical and Computer Engineering (IJECE), vol. 13, no. 3, pp. 3440–3448, Jun. 2023, doi: 10.11591/IJECE.V13I3.PP3440-3448.

J. F. Abdulkareem and H. K. Aljobouri, “Chest CT Images Analysis with Deep Learning Algorithms for COVID-19 Diagnostic for Iraqi Center,” AIP Conference Proceedings, vol. 2414, no. 1, Feb. 2023, doi: 10.1063/5.0117655/2870529.

Z. K. Alkordy, N.H., Aljobouri, H.K. and Wadi, “Feature Extraction and Selection of Kidney Ultrasound Images Using a Deep CNN and PCA,” Proceedings of 6th Computational Methods in Systems and Software 2022, vol. 1, pp. 104–114, 2023, doi: DOI: 10.1007/978-3-031-21435-6_10.

S. M. Alnedawe and H. K. Aljobouri, “A New Model Design for Combating COVID -19 Pandemic Based on SVM and CNN Approaches,” Baghdad Science Journal, vol. 20, no. 4, pp. 1402–1402, Aug. 2023, doi: 10.21123/BSJ.2023.7403.

J. Y. R. Al-Awadi, H. K. Aljobouri, and A. M. Hasan, “MRI Brain Scans Classification Using Extreme Learning Machine on LBP and GLCM,” International Journal of Online and Biomedical Engineering (iJOE), vol. 19, no. 02, pp. 134–149, Feb. 2023, doi: 10.3991/IJOE.V19I02.33987.

A. M. Hasan, N. K. N. Al-Waely, H. K. Ajobouri, R. W. Ibrahim, H. A. Jalab, and F. Meziane, “A classification model of breast masses in DCE-MRI using kinetic curves features with quantum-Raina’s polynomial based fusion,” Biomedical Signal Processing and Control, vol. 84, p. 105002, Jul. 2023, doi: 10.1016/J.BSPC.2023.105002.

A. M. Hasan, N. K. N. Al-Waely, H. K. Aljobouri, H. A. Jalab, R. W. Ibrahim, and F. Meziane, “Molecular subtypes classification of breast cancer in DCE-MRI using deep features,” Expert Systems with Applications, vol. 236, p. 121371, Feb. 2024, doi: 10.1016/J.ESWA.2023.121371.

A. M. Hasan, H. K. Aljobouri, N. K. N. Al-Waely, R. W. Ibrahim, H. A. Jalab, and F. Meziane, “Diagnosis of breast cancer based on hybrid features extraction in dynamic contrast enhanced magnetic resonance imaging,” Neural Computing and Applications, pp. 1–14, Aug. 2023, doi: 10.1007/S00521-023-08909-Y/METRICS.

N. B. Khalaf, H. K. Aljobouri, and M. S. Najim, “Identification and Classification of Retinal Diseases by Using Deep Learning Models,” in 2023 International Conference on Smart Applications, Communications and Networking, SmartNets 2023, 2023, doi: 10.1109/SmartNets58706.2023.10215740.

However, there is no need to cite the paper.

---

## [Author Response · Author response to Decision Letter 1]

17 Oct 2024

Academic editor’s comments:

Journal Requirements:

Author response: We have gone through the references and have not found any papers that have been retracted. We have updated the References list to be more consistent and added a few more references. These references are number [3], [14] and [20] in the manuscript.

Additional Editor Comments:

The number of references is very small, we prefer to add more.

Author response: We disagree with the comment regarding the low number of references. The 31 references in the original research article are adequate, and adding more references does not diminish the impact of this article. However, as the editor has requested more references, we have added three additional references to the manuscript.

---

## [Editor Report · Decision Letter 2]

21 Oct 2024

DLLabelsCT: Annotation tool using deep transfer learning to assist in creating new datasets from abdominal computed tomography scans, case study: pancreas

PONE-D-24-21753R2

Dear Dr. Mustonen,

We’re pleased to inform you that your manuscript has been judged scientifically suitable for publication and will be formally accepted for publication once it meets all outstanding technical requirements.

Kind regards,

Hadeel K. Aljobouri

Academic Editor

PLOS ONE
---

## [Editor Report · Acceptance letter]

20 Nov 2024

PONE-D-24-21753R2 

PLOS ONE

Dear Dr. Mustonen, 

I'm pleased to inform you that your manuscript has been deemed suitable for publication in PLOS ONE. Congratulations! Your manuscript is now being handed over to our production team.

Kind regards, 

on behalf of

Asst.Prof.Dr. Hadeel K. Aljobouri 

Academic Editor

PLOS ONE